# Acupuncture for Depression: A Systematic Review and Meta-Analysis

**DOI:** 10.3390/jcm8081140

**Published:** 2019-07-31

**Authors:** Mike Armour, Caroline A. Smith, Li-Qiong Wang, Dhevaksha Naidoo, Guo-Yan Yang, Hugh MacPherson, Myeong Soo Lee, Phillipa Hay

**Affiliations:** 1NICM Health Research Institute, Western Sydney University, Penrith, NSW 2751, Australia; 2Translational Health Research Institute, School of Medicine, Western Sydney University, Penrith, NSW 2751, Australia; 3School of Acupuncture and Moxibustion, Beijing University of Chinese Medicine, Beijing 100029, China; 4Department of Health Sciences, University of York, Heslington, York YO10 5DD, UK; 5Clinical Medicine Division, Korea Institute of Oriental Medicine, Daejeon 34054, Korea; 6Campbelltown Hospital, South West Sydney Local Health District, Campbelltown, NSW 5074, Australia

**Keywords:** depression, acupuncture, dosage, frequency

## Abstract

Background: Depression is commonly treated with anti-depressant medication and/or psychological interventions. Patients with depression are common users of complementary therapies, such as acupuncture, either as a replacement for, or adjunct to, their conventional treatments. This systematic review and meta-analysis examined the effectiveness of acupuncture in major depressive disorder. Methods: A search of English (Medline, PsychINFO, Google Scholar, and CINAL), Chinese (China National Knowledge Infrastructure Database (CNKI) and Wanfang Database), and Korean databases was undertaken from 1980 to November 2018 for clinical trials using manual, electro, or laser acupuncture. Results: Twenty-nine studies including 2268 participants were eligible and included in the meta-analysis. Twenty-two trials were undertaken in China and seven outside of China. Acupuncture showed clinically significant reductions in the severity of depression compared to usual care (Hedges (g) = 0.41, 95% confidence interval (CI) 0.18 to 0.63), sham acupuncture (g = 0.55, 95% CI 0.31 to 0.79), and as an adjunct to anti-depressant medication (g = 0.84, 95% CI 0.61 to 1.07). A significant correlation between an increase in the number of acupuncture treatments delivered and reduction in the severity of depression (*p* = 0.015) was found. Limitations: The majority of the included trials were at a high risk of bias for performance blinding. The applicability of findings in Chinese populations to other populations is unclear, due to the use of a higher treatment frequency and number of treatments in China. The majority of trials did not report any post-trial follow-up and safety reporting was poor. Conclusions: Acupuncture may be a suitable adjunct to usual care and standard anti-depressant medication.

## 1. Introduction

Clinical depression is characterized by behavioral, cognitive, and emotional features and is recognised as a major public health problem that has a substantial impact on individuals and on society. Depressive disorders are common in the general population. In the United States, the lifetime prevalence of a major depressive disorder (MDD) has been reported at 20.6% [1]. Most depressed patients are treated in primary care and do not require hospitalisation. In primary care, guidelines suggest depression is primarily managed with antidepressants [2]. A range of psychological interventions, including cognitive behavioural therapies, interpersonal therapy, psychotherapy, and counseling, are also recommended treatment options [3]. When patients fail to respond to a single modality, adjunctive or combined pharmacological and/or psychological treatments are recommended. However, around 30% may remain non responsive or partially responsive, even with augmentation [2].

Alternative approaches, such as complementary and alternative medicine (CAM), are frequently used in people with depression [4,5,6], either as an adjunct or replacement [4] for conventional therapies.

Contemporary acupuncture practice is commonly undertaken as part of the medical hospital system in modern China [7] and with provision either in hospital or in private practice in the United Kingdom [8]. Traditionally, acupuncture involves the insertion of fine needles into specific points, called acupuncture points, on the body to achieve a therapeutic effect. Following insertion, needles can be stimulated by hand (called ‘manual acupuncture’), or via the application of a small electrical current (called ‘electro-acupuncture’). A modern alternative is laser acupuncture, a non-penetrative form of acupuncture that uses low-power laser light to stimulate acupuncture points.

Current models of depression suggest that changes in the hypothalamic–pituitary–adrenal (HPA) axis, dysfunction among stress hormones, and disequilibrium in neurotransmitters, such as noradrenaline, serotonin, and dopamine, may be key factors in the onset and maintenance of major depressive disorder [9]. Results from animal experiments suggest a multitarget antidepressant effect of acupuncture, which may be related to amino acid metabolism and inflammatory pathways, especially the Toll-like receptor signaling pathway, tumor necrosis factor (TNF) signaling pathway, and nuclear factor kappa light chain enhancer of activated B cells (NF-kappa B) signaling pathway [10]. In addition, similar to antidepressant medications, acupuncture is capable of affecting the neurotransmitter levels of serotonin and noradrenaline, along with the adenylate cyclase cyclic adenosine monophosphate-protein kinase A (AC-cAMPPKA) cascade within the central nervous system [11].

A ‘dose’ of acupuncture is made up of multiple components. The exact components of a dose differ slightly between authors but consist of: (a) a neurophysiological dose, which includes the number of needles used, the retention time, and the type of stimulation used on the needles, and (b) a cumulative dose, made up of the frequency and total number of treatments [12,13,14]. Previous research has shown that changing the components of the acupuncture ‘dose’ appears to change the therapeutic outcomes in women undergoing in-vitro fertilization (IVF) [15] and period pain [12,16], with functional MRI (fMRI) studies showing that differing doses modulate brain activity differently [17]. Our recent Cochrane review on acupuncture for the treatment of depression [18] showed some evidence of benefits but did not explore the contribution of different dose components on the clinical outcomes for depression, for example, the number and frequency of acupuncture sessions provided.

The aim of this study was to examine the effectiveness of acupuncture compared to usual care (treatment as usual), compared to sham or placebo acupuncture, to a psychological intervention, and as an adjunct to selective serotonin reuptake inhibitors (SSRI) or serotonin and norepinephrine reuptake inhibitors (SNRI) medication. Based on a careful review of the existing literature, this study appears to be the first to explore the effect of variations in common dosage components on depression-related outcomes.

## 2. Methods

### 2.1. Search Strategy

The databases searched included:
English: Medline, PsychINFO, Google Scholar, and CINAL.Chinese: China National Knowledge Infrastructure Database (CNKI) and Wanfang Database.Korean: the Korean Studies Information Service System (KISS), DBPIA, Korea Institute of Science and Technology Information, Research Information Service System (RISS), Korea Med, Korean Medical Database (KM base), and Oriental Medicine Advanced Searching Integrated System (OASIS).

All databases were searched from 1 January 1980 to the end of November 2018, using the keywords (acupunct* or acupress* or acupoints* or electroacupunct* or electro-acupunct* or auriculotherap* or auriculoacupunct* or moxibust*) and (depress* or “affective disorder*” or “affective symptoms” or mood) using the Boolean ‘AND/OR’ operators. These keywords were the same as our recent Cochrane review of depression [18]. Both MESH and Non-MESH terms were included in this search. Papers that either had English full text, Korean full text, or Chinese full text available were eligible. Reference lists of full text papers were searched, and any relevant articles identified were screened. Figure 1 outlines the search process.

### 2.2. Eligibility Criteria

We included participants with clinical depression. Depression needed to be the primary condition rather than a co-morbidity. We included studies of people of any gender and of any ethnicity, aged 16 years or above, with clinically diagnosed depression, where depression was diagnosed via one or more of the following of the following criteria: depression defined by the Diagnostic and Statistical Manual (DSM-III, DSM-IV or DSM 5) [19,20,21], the International Classification of Disease (ICD-10) [22], the Criteria for Classification and Diagnosis of Mental Diseases (CCMD-2 or CCMD-3) [23,24], the Clinical Interview Schedule—Revised [25], or using the Beck Depression Inventory [26].

For the purposes of this review, eligible interventions comprised of manual acupuncture, electro acupuncture, or laser acupuncture. Eligible comparator groups were:
Usual care: This varies depending on geographical location and patient preference and for many participants is likely to incorporate SSRI or SNRI classes of anti-depressant medication;Sham acupuncture or placebo acupuncture, which could consist of any of the following:
Invasive acupuncture control: This includes the insertion of acupuncture needles into either ‘non-acupuncture points’, or acupuncture points that are assumed to be unrelated to the treatment of depression. Needles may be inserted superficially or to regular depths;Non-invasive acupuncture control: This could consist of non-penetrating needle devices, such as Park [27] or Streitberger [28] devices, or a sham laser device where the laser is not activated;Acupuncture plus SSRI/SNRI medication: To reflect clinical practice in western countries, where clinical guidelines are unlikely to recommend acupuncture alone as an alternative to SSRI/SNRI medication or to recommend older tri-cyclic antidepressants (TCAs) [2,3], only trials comparing acupuncture as an adjunct to SSRI or SNRI medication were eligible. For clinical relevance, trials using medication that is no longer in use (such as Flupentixol/melitracen) were not eligible;Psychological intervention, such as cognitive behavioural therapy (CBT), psychotherapy, or counseling.

For inclusion, outcome measures needed to include either a validated clinician rated measure of depression severity (such as the Hamilton Rating Scale for Depression (HAMD)) or a participant-reported measure, such as the Beck Depression Inventory (BDI) or Patient Health Questionnaire (PHQ-9). Cross over trials were excluded due to uncertainty regarding the period to allow for a washout for acupuncture treatment.

### 2.3. Data Extraction

Two authors extracted the data independently and a third author (MA or CS) resolved any disagreement. Where data was missing or unclear, the study authors were contacted via email by the authors to request the missing data be provided. Authors were contacted twice over a six-week period. If no response was received in that time, the data was marked as missing. One author (HM) was an author on an included study. HM had no part in the screening or data extraction of this study.

Data were extracted on all of the following outcomes (if reported):
-Severity of depression at the end of the intervention;-Severity of depression at follow-up (short term, medium term, and long term);-Adverse events.

Where there were two or more ‘sham’ arms used in a single study, where one or more arms were ‘invasive’, such as using minimal needling of acupuncture points, then the non-invasive arm (e.g., using a park device) was chosen, as it was expected to have less activity compared to invasive needling [29].

When multiple data points were available post intervention (e.g., end of intervention or one month follow up), end of intervention data were used as the primary time point. If follow-up data were available, they were categorised into short term (less than or equal to three months), medium term (over three and up to six months), and long term (over six months). Where there were multiple eligible intervention groups, data were combined from active treatment arms into one group, as per the Cochrane guidelines [30]. When multiple scales were used for the severity of depression, if both clinician and participant reported scales were present, priority was given to clinician reported outcomes.

### 2.4. Risk of Bias

Included studies were also assessed using the Cochrane Collaboration’s risk-of-bias tool [30], as per our previous review [31]. Any disagreements were resolved by discussion.

### 2.5. Acupuncture Quality

The quality of the acupuncture delivered in the included trials was assessed using the National Institute for Complementary Medicine Acupuncture Network (NICMAN) scale. [32] This recently developed tool is reliable and is recommended to be used in conjunction with the Standards for Reporting Interventions in Clinical Trials of Acupuncture (STRICTA) checklist [33]. The NICMAN consists of 11 items and is scored out of 23, with four main domains of appraisal:
(1)Population Intervention Comparator Outcome measure (PICO) reporting, study design relevance, and paradigm based differential diagnosis (items 1–6);(2)Acupuncture point selections and locations, in reference to published literature (items 7 and 9);(3)Description of the needle dimensions, needling technique, and number of treatments (items 8 and 10);(4)Acupuncturist qualification and training (item 11).

Higher scores on the NICMAN scale are associated with an increase in the quality of the acupuncture intervention delivered. NICMAN scores were assessed by two authors independently, with any disagreement resolved by a third author (MA or CS).

### 2.6. Meta-Analysis

Random-effects meta-analyses were conducted using the Comprehensive Meta-Analysis software (Version 3). Intervention effect sizes were pre-post changes between intervention and control groups for the primary outcome measure (change in severity of depression) and were calculated using Hedges’ g statistic [34], along with 95% confidence intervals (CIs) around the estimated effect-size. For more details, please refer to our previous study [31].

### 2.7. Subgroup Analysis

The following sub-group analyses were decided on *a priori:*-Comparison between different acupuncture subtypes (manual, electro-acupuncture, and laser acupuncture)*;-Needle retention time* (<20 min versus >20 min);-Fixed versus individualised acupuncture.

Those sub-group analyses marked with a * indicate that these are components of acupuncture ‘dosage’. *A postori,* a decision was made to examine if cultural or geographical factors might influence the response to acupuncture treatment, so a sub-group analysis between studies conducted in China versus countries other than China was undertaken.

Variations in dose components (such as total number of treatments) between China compared to countries other than China will be explored via *t*-test for normally distributed data or a Mann–Whitney U test for non-normally distributed data.

If the frequency of treatment varied over the course of treatment (e.g., twice per week for the first week, then once per week for eight weeks), the frequency used in analysis was the average frequency calculated over the course of treatment. For studies that reported a treatment frequency of every second day, a treatment frequency of 3.5 treatments per week was used in the analysis. In studies where there was semi-standardised or individualised treatment, the number of acupuncture points was determined by either the minimum number that was required by the study protocol (excluding any optional points), or the average number of points used by practitioners, if this was reported in the manuscript. If both were reported, then preference was given to the average number of points used.

The relationship between continuous moderators and estimates of effect size were explored with a meta-regression analysis. These were performed for characteristics of acupuncture dosage that may have had an impact on the severity of depression—the frequency of treatment, the total number of treatments, the number of acupuncture points used—and for the quality of the acupuncture delivered, determined by the NICMAN scale score.

## 3. Results

A total of 29 studies including 2268 participants were included in this review. Table 1 outlines the characteristics of the included studies.

### 3.1. Design

A parallel group design was used in all trials. Twenty-six trials had two eligible groups (acupuncture plus a control group). Three trials [35,36,37] had three eligible arms, where the unit of analysis was adjusted based on the description in the methods.

### 3.2. Control Groups

Control groups varied between studies dependent on the research question. Sixteen trials [35,37,38,39,40,41,42,43,44,45,46,47,48,49,50,51] used SSRI or SNRI medication alone as the control. Twelve trials used an acupuncture control, with nine trials using sham acupuncture [52,53,54,55,56,57,58,59,60,61] and two trials [62,63] using an invasive control. One trial [36] of acupuncture plus usual care had two control groups, one of counseling plus usual care and one usual care alone.

### 3.3. Country

Twenty-two studies were undertaken in Chinese speaking countries (20 in mainland China and two in Hong Kong), and seven in English speaking countries (three in the United States of America, two in Australia, and two in the United Kingdom).

### 3.4. Sample Sizes

Sample sizes of trials included in this review ranged from 19 [59] to 755 [36], with an overall median of 60. Trials undertaken in Chinese speaking countries had a median sample size of 61, while those in English speaking countries had a median sample size of 37.

### 3.5. Participant Criteria

Trials recruited participants who met the diagnostic criteria for depression or who had a clinical presentation of depression, as defined by the trial authors. All trials recruited participants with major depressive disorder. Seven trials used the Diagnostic and Statistical Manual of Mental Disorders (DSM-II, III, IV or V), 11 trials used the Chinese Classification of Mental Disorders (CCMD-2 or 3), six trials used the Criteria for Diagnosis of Depression (ICD-10), two trials used the Beck Depression Inventory (BDI), and one trial used the Clinical Interview Schedule-Revised (CIS-R). One trial used both the CCMD-3 and ICD-10 [41], and one trial used both the CCMD-3 and DSM-IV [61].

### 3.6. Interventions

The acupuncture delivered in the trials varied in terms of point selection, frequency of treatments, and total number of treatments administered.

#### 3.6.1. Type of Acupuncture

Eighteen trials used manual acupuncture, six trials used electro acupuncture, two trials used laser acupuncture, two trials included both manual and electro acupuncture arms, and one trial combined electro and manual acupuncture.

#### 3.6.2. Frequency of Treatment

The frequency of acupuncture treatment ranged from daily to approximately once per week. Overall, 19 trials used acupuncture treatment three times a week or more, three trials used acupuncture treatment twice per week [52,55,56], five trials used acupuncture treatment once a week on average [36,57,58,62,63]. The frequency of acupuncture treatment was unclear in two trials [54,59]. In English speaking countries, the median frequency of treatments was once per week, while in Chinese speaking countries, the median frequency was treatment every second day (3.5 treatments per week). The median frequency of treatments delivered in China was significantly greater (U = 1, *p* < 0.00061). 

#### 3.6.3. Total Number of Treatments

The total number of acupuncture treatments ranged from nine treatments [53,60] to 40 treatments [42], with a median of 18 treatments, in the included studies. The total number of acupuncture treatments was unclear in one trial [54]. All trials in English speaking countries delivered 12 treatments, with trials undertaken in Chinese speaking countries delivering a median of 24 treatments. The median number of treatments delivered in China was significantly greater (U = 14, *p* = 0.0006).

#### 3.6.4. Duration

The total duration of acupuncture treatments ranged from three weeks [53,60] to 12 weeks [36,55,56], with a median treatment duration of six weeks, in the included studies. The total duration of treatment was unclear in one trial [59].

#### 3.6.5. Treatment Protocol

Fifteen trials used a standardised treatment protocol, with a fixed selection of points administered at each acupuncture session. The selection of points varied and included acupuncture points located on the arms, legs, abdomen, and head. Eleven trials used a semi-standardised treatment protocol consisting of semi-fixed points, including a pre-defined set of points used in combination with points selected on the basis of diagnosis and symptomatic patterns. Three trials used an individualised treatment protocol, with individual points selected on the basis of diagnosis and symptomatic patterns. Of the 15 trials using fixed points, the number of acupuncture points ranged from ten points [61] to two points [35,52], with a median number of six points. The number of acupuncture points was not reported in one of the trials using fixed points [62].

#### 3.6.6. Needle Retention Times

The needle retention times ranged from 20 min [59,63] to 60 min [38,39]. Overall, 20 trials used a needle retention time of 30 min or less, and four trials used a needle retention time of more than 30 min. In five trials, the needle retention time was determined as unclear.

#### 3.6.7. NICMAN Scores

NICMAN scores ranged from 7 [44,54] to 23 [63]. The median NICMAN score was 18.

### 3.7. Outcome Measures

Twenty-seven trials assessed the primary outcome of depression using the Hamilton Rating Scale for Depression (HAMD) [64], one trial [37] used the Beck Depression Inventory (BDI) [26], and one trial [36] used the PHQ-9 [65]. Only two trials [36,61] collected data for post-treatment follow-up, all others collected data to the end of the intervention. Seven studies reported adverse events. Four studies reported adverse event counts only [52,59,60,66], two used the Asberg Side Effect Rating Scale (SERS) scale and one used the Toxic Exposure Surveillance System (TESS) [38].

### 3.8. Risk of Bias in Included Studies

See Figure 2 for a graphical summary of risk of bias assessments performed by review authors for the 29 included studies, based on the seven risk-of-bias domains. No trials were assessed as having a low risk-of-bias across all domains. Three trials were at low risk-of-bias for six domains [52,53,57] and four trials were at low risk-of-bias for five domains [36,46,58,60]

#### 3.8.1. Randomization

Twenty-six trials were at a low risk of bias for randomization and three trials did not describe the method of randomization or were assessed as unclear [48,59,63].

#### 3.8.2. Allocation

Fourteen trials were assessed as low risk of bias for allocation concealment. The remaining 15 trials did not report the method of allocation, so we assessed their risk as unclear.

#### 3.8.3. Blinding

We assessed blinding as providing a low risk of performance bias in ten trials. For studies comparing acupuncture versus a sham or placebo acupuncture control, we sought evidence of verification of the blinding of participants, such as tests of blinding being reported. Most trials involved comparisons of acupuncture versus usual care or medication and participants could not be blinded; this contributed to an assessment of high risk. We assessed a total of 18 trials as having a high risk for performance bias and one trial as being unclear.

Most trials used clinician rated outcome measures. Ten trials were rated as having a low risk of detection bias, nine trials as high risk and the remaining ten did not report on blinding of the assessor/clinician, the analyst, or outcome measures, and were rated as unclear risk.

#### 3.8.4. Incomplete Outcome Data

We assessed the majority of trials as having a low risk of bias for outcome reporting. Eight trials were at high risk, owing to dropout rates or incomplete data, and attrition bias was unclear in six trials. We rated trials as having high risk of bias if dropout rates were uneven between groups and the reason for dropout was related or suspected to be related to group allocation. We also rated trials as having a high risk for bias if investigators reported a dropout rate >20% and did not report how they dealt with this (e.g., intention to treat (ITT) analysis, last observation carried forward).

#### 3.8.5. Selective Reporting

Twenty-six trials were rated as unclear risk, owing to no study protocol or trial registration records being available. Two trials [36,57] did report data on all included outcomes and were at low risk of bias. We rated one trial as having a high risk of bias [46].

#### 3.8.6. Other Potential Sources of Bias

Risk of bias was unclear for nine trials. We rated risk from other sources of bias as low for 12 trials. We assessed an imbalance at randomisation in eight trials, and these were rated as high risk of bias [41,43,50,55,59,61,67].

### 3.9. Acupuncture versus Usual Care

One study [36] compared acupuncture plus usual care to usual care alone. There was evidence of a moderate benefit in the severity of depression in the acupuncture group compared to usual care g = 0.41 (N = 1, *n* = 301, 95% CI 0.18 to 0.63).

### 3.10. Acupuncture versus Sham

The pooled effect of acupuncture on the severity of depression compared to sham at the end of the intervention (Figure 3) was moderate g = 0.55 (N = 12, *n* = 646, 95% CI 0.31 to 0.79, Q = 38.15, *p* < 0.001, I^2^ = 71.2). When comparing different types of acupuncture, both manual (g = 0.96, N = 7, *n* = 439, 95% CI 0.50 to 1.4, Q = 25.1, *p* < 0.001, I^2^ = 76.1) and laser (g = 0.90, N = 2, *n* = 63, 95% CI 0.37 to 1.4, Q = 235, *p* = 0.62, I^2^ = 0) showed large effect sizes, while electro-acupuncture did not show a significant benefit compared to sham (g = 0.20, N = 3, *n* = 144, 95% CI −0.13 to 0.53, Q = 295, *p* = 0.863, I^2^ = 0). Only one study [61] contributed data for short term follow-up, so no meta-analysis was performed on follow-up data.

### 3.11. Acupuncture Plus SSRI/SNRI versus SSRI/SNRI Alone

The pooled effect of acupuncture plus SSRI/SNRI medication on the severity of depression at the end of intervention compared to SSRI/SNRI medication alone (Figure 4) was large g = 0.84 (N = 18, *n* = 1169, 95% CI 0.61 to 1.07, Q = 87, I^2^ = 80.5). When comparing different types of acupuncture in combination with SSRI/SNRI medication, both manual (g = 1.1, N = 12, *n* = 865, 95% CI 0.74 to 1.49, Q = 74, I^2^ = 85) and electro-acupuncture (g = 0.85, N = 5, *n* = 264, 95% CI 0.52 to 1.17, Q = 5.9, I^2^ = 32.4) showed large effect sizes, while the single study using combined manual and electro-acupuncture showed no evidence of significant benefits (g = 0.02, N = 1, *n* = 40, 95% CI −0.61 to 0.65).

### 3.12. Acupuncture versus Psychological Intervention

Only one study [36] compared acupuncture plus usual care to counseling plus usual care. There was no evidence of a significant difference in the severity of depression between groups (g = 0.12, N = 1, *n* = 453, 95% CI −0.07 to 0.32).

### 3.13. Adverse Events

When compared to sham, there was no increased rate of adverse events (RR1.66, N = 3, *n* = 146, 95% CI 0.56 to 4.9, Q = 2.6, I^2^ = 25). When compared to SSRI medication alone, there was no difference in adverse event scores (SMD 1.17, N = 3, *n* = 171, 95% CI −0.53 to 2.89, Q = 2, I^2^ = 95.8).

### 3.14. Subgroup Analysis

There was no difference in the reduction in severity of depression when comparing fixed, semi-standardised, and individualised acupuncture sub-groups (*p* = 0.468), or when comparing different needle retention times (*p* = 0.796). There were not enough studies using participant reported outcomes to allow meaningful comparison with clinician reported scales.

Trials in China had large effects in the reduction in severity of depression (g = 0.96, N = 24, *n* = 1599, 95% CI 0.72 to 1.2, Q = 114, I^2^ = 79.9), whereas trials undertaken outside of China had only small effects (g = 0.38, N = 7, n = 970, 95% CI 0.15 to 0.60, Q = 12, I^2^ = 45) (Figure 5). A test for between group differences showed that trials from China had significantly greater reductions in the severity of depression compared to those undertaken outside of China (Q = 12, *p* < 0.001).

The meta-regression analysis (Figure 6) found that a greater total number of treatments was related to a greater reduction in the severity of depression (N = 29, *n* = 2268, *B* = 0.025, SE = 0.010, *Z* = 2.44, *p* = 0.015). Meta-regression showed a trend towards a greater reduction in the severity of depression with more frequent treatment (N = 29, *n* = 2268, *B* = 0.095, SE = 0.050, *Z* = 1.88, *p* = 0.061).

A higher NICMAN score did not have a significant association with any difference between groups in the reduction in severity of depression (N = 29, *n* = 2268, *B* = −0.011, SE = 0.033, *Z* = −0.33, *p* = 0.73). The reporting on the number of acupuncture points used was only provided in 18 studies. There was no significant association between the number of acupuncture points and the reduction in the severity of depression (N = 29, *n* = 2268, *B* = 0.055, SE = 0.079, *Z* = 0.70, *p* = 0.48).

### 3.15. Publication Bias

There was no evidence of publication bias (*p* = 0.079) according to the Begg and Mazumdar test. Appendix A shows the funnel plot for publication bias. A Duval and Tweedie trim and fill analysis did not result in any studies to the left of the mean being trimmed.

## 4. Discussion

### 4.1. Principal Findings

In this review we identified 29 studies of acupuncture for depression, and these studies included 2268 participants. Acupuncture was found to be associated with clinically relevant effect sizes in reducing the severity of depression when compared to usual care alone, to sham acupuncture, and to SSRI/SNRI medication alone. With regard to the usual care comparison, the one relevant trial found moderate statistically significant benefits for those receiving acupuncture, with an effect size of 0.41. This was by far the largest trial, with 755 patients, and the only one to address the practical question of what impact acupuncture might have on patients with depression in primary care, who are routinely referred to acupuncturists. With regard to the sham acupuncture comparison, our meta-analysis of 12 trials showed acupuncture to significantly outperform varied forms of sham interventions, with a moderate effect size of 0.55, suggesting that acupuncture is more than a placebo. With regard to the comparison of acupuncture plus SSRI anti-depressants versus SSRI anti-depressants alone, our meta-analysis of 17 trials showed a moderate to large, and significant, effect size of 0.83, suggesting a potential role for acupuncture to be more routinely provided as an adjunct to SSRIs in cases where SSRIs are found to be ineffective or poorly tolerated.

Our review is the first to explore the correlation between the total number of treatments and the effect size, with the finding that when delivering acupuncture in a clinical setting, there appears to be benefits in longer courses of treatment. We found a trend between increased treatment frequency and a reduction in the severity of depression. The number of trials that delivered less than two treatments per week or more than five was small, and the addition of more studies with these treatment frequencies may change this result. Overall, it appears that the total number of treatments plays a greater role in reduction of depression than does the frequency of delivery of those treatments.

### 4.2. Strengths and Limitations

This review and meta-analysis had several strengths. The use of English, Chinese, and Korean databases allowed access to publications in the three most common languages for acupuncture research. The majority of included studies used clinician outcome measures to measure the severity of depression, and where multiple outcomes were given, clinician rated scales were given preference. Compared to the most recent Cochrane review of acupuncture for depression [18], within which databases were searched only up to June 2016, the present review differs in several important aspects. This review includes studies published until December 2018, increases clinical relevance by including only patients with depression as the primary presentation, includes a quality assessment measure for the acupuncture delivered in the form of the NICMAN scale, and explores the contribution of various dosage components on depression related clinical outcomes.

In terms of the limitations of this meta-analysis, the benefits we observed only relate to outcomes at the end of treatment. There was insufficient evidence to determine if the benefits were sustained due to the small number of trials (*n* = 2) that undertook any post trial follow-up. A key caveat with regard to the overall findings is the level of risk of bias across the seven risk of bias domains. No trials were assessed as having low risk of bias across all domains. Only four trials were identified as having a low risk of bias across at least five of these domains. Moreover, the transferability of the results may be compromised by the preponderance of trials from China. Only seven of the 29 trials were conducted in other countries, namely USA, UK, and Australia. The effect sizes were larger in China, which may be explained by the significantly more frequent treatment regimen and the significantly greater total number of sessions. The median number of total treatments delivered was twice that in China compared to other countries, and the median frequency of treatment much greater, with treatment being delivered with a median frequency of every second day in China, compared to once per week. A high risk of bias is associated with larger effect sizes [68], therefore an alternate view is that the high risk of bias of the trials in China may explain the larger effects seen. The percentage of trials assessed as having a low risk of bias in five or more domains conducted in China was 14% (3/22), while in other countries the rate was 57% (4/7).

The present review found moderate to large effect sizes when adding acupuncture to first line antidepressants, such as SSRIs. People who favour alternative treatments may elect to have acupuncture rather than other adjunctive approaches, such as a mood stabilizing agent like lithium. However, limitations of the research involving acupuncture and anti-depressants, particularly the high risk of bias from lack of blinding, tempers this conclusion. It should also be noted that all the trials involving acupuncture compared to SSRIs alone were conducted in China. Moreover, blinding for those participants in the acupuncture plus medication versus medication alone comparisons was not performed. This lack of blinding is likely to increase the non-specific/attention effects of those in the treatment group and may result in overestimating the effect size. Similarly, reporting on the blinding status of those clinicians who undertook the rating for depression was often unclear and this also may overestimate the effect size.

### 4.3. Implications for Research and Practice

In terms of the implications for research, the stand-out requirement is for more higher-quality randomised control trials to evaluate the effectiveness of acupuncture for treatment for depression. Such trials need to be designed with a low risk of bias, with large enough sample sizes to be sufficiently powered, with long enough follow-ups beyond the end of treatment, ideally for 12 months after randomisation, and with sufficient numbers of treatment sessions provided. More research into the dosage of acupuncture would be helpful, with the intention of clarifying what comprises a sufficient number of acupuncture sessions for the average patient. In terms of trial designs, this review has highlighted the need for more practical trials, in which the treatment as usual, or usual care, is the primary comparator, as only one trial [36] included a usual care comparison. More trials of this type are needed to model real-world settings, and in different countries. The results of these trials will provide important evidence for patients when considering the option of acupuncture, for clinicians when considering the referral of their patients to an acupuncturist, for decision-makers when considering providing resources to fund mental health treatments, and for policy-makers when considering questions of health policy related to depression. Moreover, such pragmatic designs provide an ideal framework for a parallel cost-effectiveness analysis, thereby supporting economic-related decision-making.

In terms of implications for clinical practice, the findings of this review provide some limited support for acupuncture as a treatment for depression. Most patients with depression will be offered SSRI or SNRI anti-depressants as a first line treatment, and our evidence shows a moderate effect size is associated with acupuncture provided for this patient group. A smaller effect size in a single trial was found when acupuncture plus usual care was compared to usual care alone. The difference may in part be explained by the former trials all being conducted in China, where significantly more treatment sessions were delivered, and the latter trial recruiting from primary care in the UK, with only those patients who had moderate to severe depression, and who mostly received only one treatment a week. There remain many unanswered questions regarding what might be considered optimal acupuncture, including the number of sessions, frequency of sessions, diagnostic frameworks, flexibility of point prescriptions, role of theoretically informed life-style advice, and the engagement and self-efficacy of the patient. Manual acupuncture showed larger effect sizes than electro-acupuncture when compared to both sham and SSRI/SNRI medication alone. This is an unexpected finding because compared to manual acupuncture, electro-acupuncture has been thought to deliver a greater ’dose’ of acupuncture, as a result of the duration and intensity of stimulation [69]. Similar to our findings in this regard, evidence for additional benefits of adding electro-stimulation to acupuncture is unclear when outcomes are not related to pain [69,70].

There remain further unanswered questions regarding which patients might find acupuncture most beneficial, in terms of severity of depression, whether on anti-depressants or not, and the implications of comorbidity. We know that when a patient is depressed, they often are experiencing a range of other symptoms, commonly chronic pain, insomnia, digestive disorders, and low energy. Typically, a traditional acupuncturist will take into account this range of symptoms when constructing a treatment plan [71]. In cases where multi-morbidity from the perspective of biomedicine is perceived as problematic, research is needed to confirm, or otherwise, whether a multi-faceted acupuncture treatment provides a useful single modality solution.

## 5. Conclusions

Acupuncture showed clinically-relevant benefits in reducing the severity of depression at the end of treatment compared to sham acupuncture, usual care, and as an adjunct treatment to anti-depressant medication. The high risk of bias for participant blinding in the majority of studies means these results must be interpreted cautiously. Both the total number of treatments and the frequency of treatment may play a role in depression-related outcomes. The clinical relevance of these findings to a non-Chinese population requires further interpretation, due to the majority of studies being undertaken in China, where treatment frequencies and the total number of treatments were significantly greater than those undertaken in English speaking countries, but where trials were also rated as having a higher risk of bias. Future studies need to include both short, medium-, and long-term follow-up to determine if any benefits on the severity of depression are maintained.

## Figures and Tables

**Figure 1 jcm-08-01140-f001:**
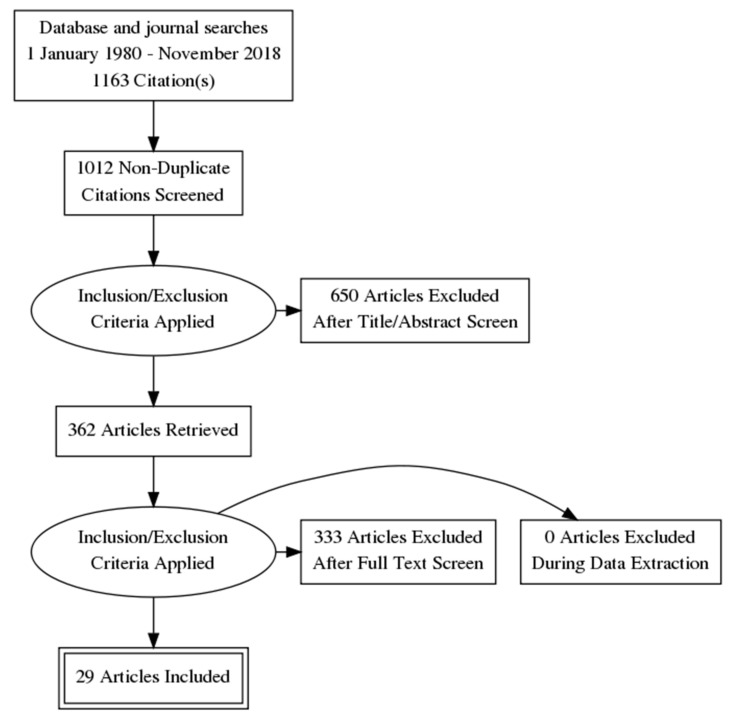
PRISMA flow diagram.

**Figure 2 jcm-08-01140-f002:**
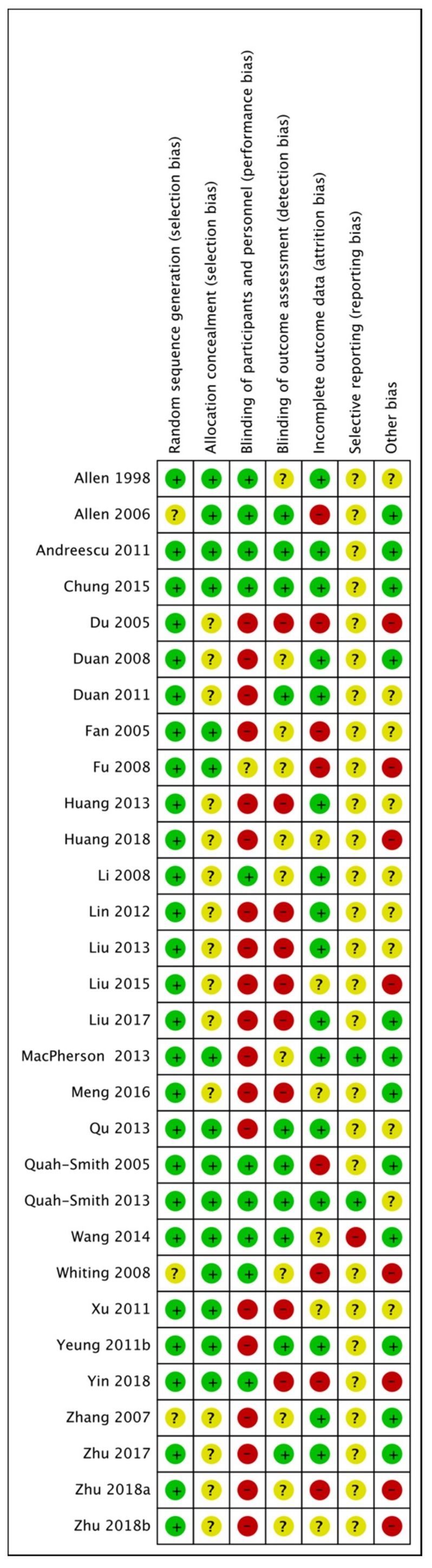
Risk-of-bias. Green, low risk of bias; yellow, unclear risk of bias; red, high risk of bias.

**Figure 3 jcm-08-01140-f003:**
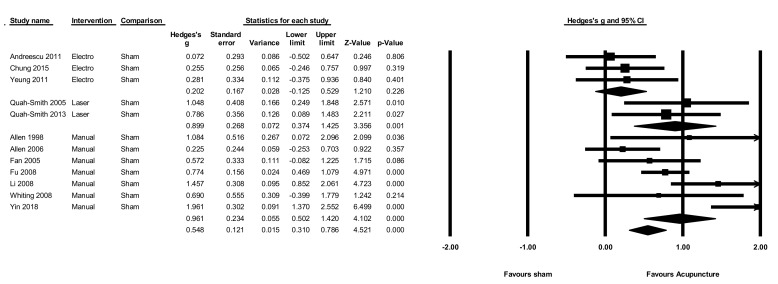
Acupuncture versus sham control on the severity of depression.

**Figure 4 jcm-08-01140-f004:**
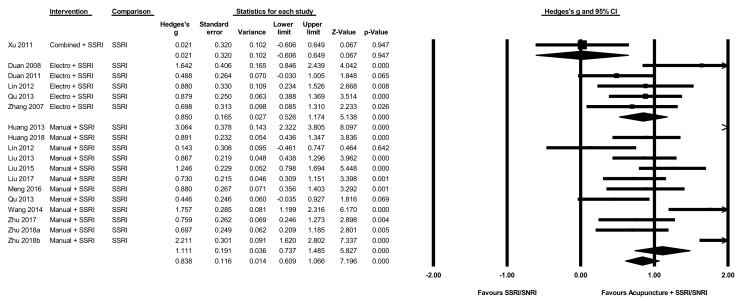
Acupuncture plus SSRI/SNRI versus SSRI/SNRI alone on the severity of depression.

**Figure 5 jcm-08-01140-f005:**
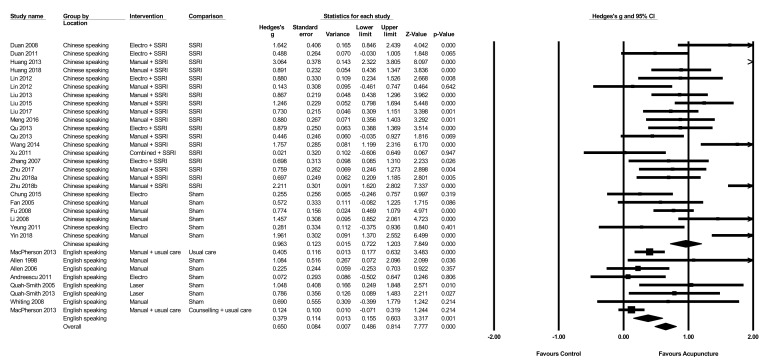
The effect of acupuncture on the severity of depression by study location.

**Figure 6 jcm-08-01140-f006:**
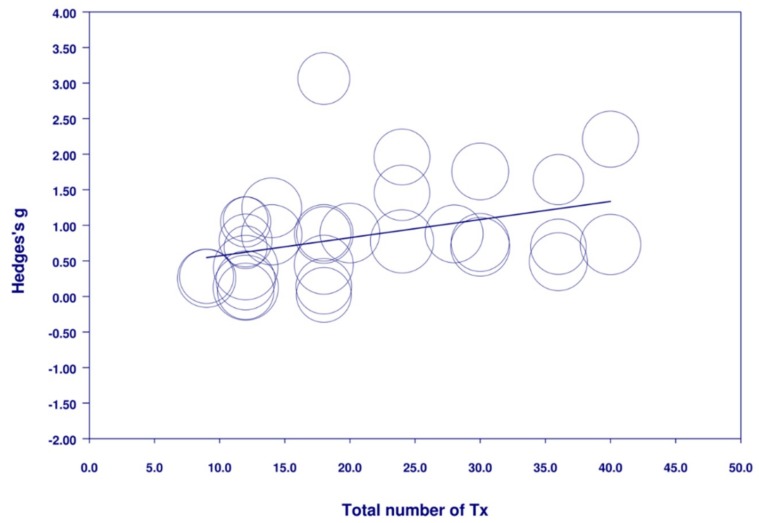
Meta-regression of total number of treatments on the severity of depression.

**Table 1 jcm-08-01140-t001:** Summary of included studies.

Study ID	Country	Sample Size	Diagnosis for Inclusion ^d^	Intervention	Control	Number of Treatments	Frequency of Treatment	Variation in Acupuncture Points	Total Number of Acupuncture Points (for Fixed Points Only)	Outcome Measures	NICMAN Score
Allen 1998 [62]	USA	17 ^a^	DSM IV	Manual acupuncture	Invasive (Non-specific)	12	2/week then 1/week	Fixed	NR ^c^	HAMD	21
Allen 2006 [63]	USA	74 ^a^	DSM-IV	Manual acupuncture	Invasive (Non-specific)	12	2/week then 1/week	Individualised	N/A	HAMD	23
Andreescu 2011 [52]	USA	45	DSM IV	Electro acupuncture	Sham	12	2/week	Fixed	2	HAMD	21
Chung 2015 [53]	Hong Kong	60 ^a^	DSM IV	Electro acupuncture	Sham	9	3/week	Fixed	4	HAMD	19
Duan 2008 [38]	China	35	CCMD	Electro acupuncture + Fluoxetine 20 mg	Fluoxetine 20 mg	36	6/week	Semi-fixed	N/A	HAMD	16
Duan 2011 [39]	China	60	ICD	Electro acupuncture + Fluoxetine 20 mg	Fluoxetine 20 mg	36	6/week	Semi-fixed	N/A	HAMD	16
Fan 2005 [54]	China	39 ^a^	CCMD-2	Manual acupuncture	Sham	NR ^c^	NR ^c^	Fixed	4	HAMD	7
Fu 2008 [55]	China	176 ^a^	CCMD-2	Manual acupuncture	Sham	24	2/week	Fixed	4	HAMD	14
Huang 2013 [40]	China	60	CCMD	Manual acupuncture + Paroxetine 20–40 mg	Paroxetine 20–40 mg	18	3–4/week	Fixed	6	HAMD	10
Huang 2018 [41]	China	80	ICD-10, CCMD-3	Manual acupuncture + Paroxetine, Sertraline, Citalopram, Fluoxetine (dosage NR ^c^)	Paroxetine, Sertraline, Citalopram, Fluoxetine (dosage NR ^c^)	20	5/week	Fixed	6	HAMD, SF-36, TESS	19
Li 2008 [56]	China	52	CCMD-2	Manual acupuncture	Sham	24	2/week	Fixed	4	HAMD	12
Lin 2012 [35]	China	92 ^b^	ICD	1. Electro acupuncture 2. Manual acupuncture + Paroxetine (dosage NR ^c^)	Paroxetine (dosage NR ^c^)	18	3–4/week	1. Fixed 2. Semi-fixed	1. 2 2. N/A	HAMD	11
Liu 2013 [44]	China	90	CCMD-3	Manual Acupuncture + Fluoxetine or Paroxetine (dosage NR ^c^)	Fluoxetine or Paroxetine (dosage NR ^c^)	14	3–4/week	Fixed	5	HAMD	7
Liu 2015 [43]	China	90	CCMD-3	Manual acupuncture + Fluoxetine, paroxetine, citalopram 20–60 mg, sertraline 50–200 mg or fluvoxamine 50–300 mg	Fluoxetine, paroxetine, citalopram 20–60 mg, sertraline 50–200 mg or fluvoxamine 50–300 mg	14	3–4/week	Semi-fixed	N/A	HAMD	15
Liu 2017 [42]	China	91	ICD-10	Manual acupuncture + Venlafaxine 75–225 mg	Venlafaxine 75–225 mg	40	5/week	Fixed	8	HAMD	15
MacPherson 2013 [36]	UK	755	BDI	Manual acupuncture + usual care	1.Counseling + usual care 2. Usual care	12	1/week	Individualised	N/A	PHQ-9	20
Meng 2016 [45]	China	60	CCMD-3	Manual acupuncture + Fluoxetine (2 tablets dosage NR ^c^)	Fluoxetine (2 tablets dosage NR ^c^)	28	7/week	Semi-fixed	N/A	HAMD	16
Qu 2013 [37]	China	160 ^b^	ICD	1. Electro acupuncture 2. Manual acupuncture + Paroxetine 10–40 mg	Paroxetine 10–40 mg	18	3/week	Fixed	7	BDI	19
Quah-Smith 2005 [58]	Australia	26	BDI	Laser acupuncture	Sham	12	2/week then 1/week	Individualised	N/A	HAMD	19
Quah-Smith 2013 [57]	Australia	37	DSM IV	Laser acupuncture	Sham	12	2/week then 1/week	Fixed	5	HAMD	21
Wang 2014 [46]	China	71	ICD	Manual acupuncture + Fluoxetine 20 mg, paroxetine 20 mg or duloxetine 40 mg	Fluoxetine 20 mg, paroxetine 20 mg or duloxetine 40 mg	30	5/week	Semi-fixed	N/A	HAMD	19
Whiting 2008 [59]	UK	17	CISR	Manual acupuncture	Sham	12	NR^c^	Semi-fixed	N/A	HAMD	19
Xu 2011 [47]	China	40	CCMD	Manual + Electro acupuncture combined + Citalopram, paroxetine or fluoxetine 20 mg	Citalopram, paroxetine or fluoxetine 20 mg	18	3/week	Fixed	7	HAMD	9
Yeung 2011 [60]	Hong Kong	39 ^a^	DSM-IV	Electro acupuncture	Sham	9	3/week	Fixed	8	HAMD	17
Yin 2018 [61]	China	64	CCMD-3, DSM-IV	Manual acupuncture	Sham	24	3/week	Fixed	10	HAMD	18
Zhang 2007 [48]	China	42	CCMD	Electro acupuncture + Paroxetine 10–40 mg	Paroxetine 10–40 mg	36	7/week	Semi-fixed	N/A	HAMD	14
Zhu 2017 [49]	China	61	CCMD-3	Manual acupuncture + Fluoxetine, paroxetine, fluvoxamine, sertraline, citalopram, estalcitalopram (dosage NR ^c^)	Fluoxetine, paroxetine, fluvoxamine, sertraline, citalopram, estalcitalopram (dosage NR ^c^)	30	5/week	Semi-fixed	N/A	HAMD	19
Zhu 2018 [50]	China	67	CCMD-3	Manual acupuncture + SSRI (Type and dosage NR ^c^)	SSRI (Type and dosage NR ^c^)	30	5/week	Semi-fixed	N/A	HAMD	20
Zhu 2018 [51]	China	70	ICD-10	Manual acupuncture + Sertraline (dosage NR ^c^)	Sertraline (dosage NR ^c^)	40	5/week	Semi-fixed	N/A	HAMD	21

^a^ Three-armed trial, only eligible arms included in sample size. ^b^ Three-armed trial, both intervention arms eligible and included. ^c^ NR = Not reported. ^d^ DSM = Diagnostic and Statistical Manual (DSM-III, DSM-IV or DSM 5), ICD-10 = International Classification of Disease, CCMD, CCMD-2, CCMD-3 = Criteria for Classification and Diagnosis of Mental Diseases, CISR = Clinical Interview Schedule—Revised, BDI = Beck Depression Inventory.

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
