# Peer review of "Acupuncture for Depression: A Systematic Review and Meta-Analysis"

_jcm, 2019, doi:10.3390/jcm8081140_

Round 1
Reviewer 1 Report
Excellent and relevant study. Minor grammatical changes and some areas needing further clarification. See comments saved to PDF.

Author Response
Thank you for your very helpful feedback, it is very much appreciated. We have addressed the comments you have made in the PDF as follows:
We have modified the introduction to state:
"The aim of this study was to examine acupuncture’s effectiveness compared to usual care (treatment as usual), compared to sham or placebo acupuncture, to a psychological intervention, and as an adjunct to SSRI/SNRI medication. Based on a careful review of the existing literature this study appears to be the first to explore the effect of variations in common dosage components on depression related outcomes"
Removed superfluous bracket at the end of the boolean terms.
Removed the 'or' in-between the terms:
"where depression was diagnosed via one or more of the following of the following criteria: depression defined by the Diagnostic and Statistical Manual (DSM-III, DSM-IV or DSM 5) [19-21], the International Classification of Disease (ICD-10) [22], the Criteria for Classification and Diagnosis of Mental Diseases (CCMD-2 or CCMD-3) [23, 24], the Clinical Interview Schedule—Revised [25]or using the Beck Depression Inventory [26]."
Made the justification for not including TCAs clearer
1. Acupuncture plus SSRI/SNRI medication. To reflect clinical practice in western countries where clinical guidelines are unlikely to recommend acupuncture alone as an alternative to SSRI/SNRI medication or to recommend older tri-cyclic antidepressants (TCAs) [2, 3], only trials comparing acupuncture as an adjunct to SSRI or SNRI medication were eligible. For clinical relevance trials using medication that is no longer in use (such as Flupentixol/melitracen) were not eligible.
Modified sentence to read studies instead of trials
4.1. Principal Findings
In this review we identified 29 studies of acupuncture for depression, and these studies included 2268 participants.
Reviewer 2 Report
This systematic review is a tremendous feat and will be paramount to the acupuncture evidence base for depression. Overall, excellent, well-designed research methodology. Well described strengths and limitations and further questions to be answered.
Very minor editing suggested.
-Abstract:
Can reword this sentence to make more sense, as not quite clear on what "between increase is number"?
"A significant correlation between increase is number of acupuncture treatments delivered and reduction in the severity of depression (p = 0.015) was found."
-Page 11 section 3.12 - change counselling to counseling
Page 15: 2nd paragraph: Clarify this sentence of "...who in the main received only one treatment a week." Is main meaning a country?
"The difference may in part be explained by the former trials all being conducted in China where significantly more treatment sessions were delivered and the latter trial recruiting from primary care in the UK only those patients who had moderate to severe depression, and who in the main received only one treatment a week."
Author Response
Thank you to the reviewer for your valuable feedback, we have addressed this as follows:
-Abstract:
Can reword this sentence to make more sense, as not quite clear on what "between increase is number"?
"A significant correlation between increase is number of acupuncture treatments delivered and reduction in the severity of depression (p = 0.015) was found."
Apologies this was a spelling error. It now reads:
A significant correlation between increase in number of acupuncture treatments delivered and reduction in the severity of depression (p= 0.015) was found
-Page 11 section 3.12 - change counselling to counseling
We have changed to 'counseling' in all areas of the manuscript for consistency
Page 15: 2nd paragraph: Clarify this sentence of "...who in the main received only one treatment a week." Is main meaning a country?
"The difference may in part be explained by the former trials all being conducted in China where significantly more treatment sessions were delivered and the latter trial recruiting from primary care in the UK only those patients who had moderate to severe depression, and who in the main received only one treatment a week."
We have changed this to a more common expression:
The difference may in part be explained by the former trials all being conducted in China where significantly more treatment sessions were delivered and the latter trial recruiting from primary care in the UK only those patients who had moderate to severe depression, and who mostly received only one treatment a week.